



# Impact of sky conditions on net ecosystem productivity of a "floating blanket" wetland in southwest China

Yamei Shao [1, 2], Huizhi Liu [1, 2*], Qun Du [1, 2], Yang Liu [1, 2], Jihua Sun [3], Yaohui Li [4]

[1]State Key Laboratory of Atmospheric Boundary Layer Physics and Atmospheric Chemistry, Institute of Atmospheric Physics,
Chinese Academy of Sciences, Beijing 100029, China
[2]University of Chinese Academy of Sciences, Beijing 100029, China
[3]Yunnan Meteorological Observatory, Kunming 530100, China
[4]College of Aviation Meteorology, Civil Aviation Flight University of China, Guanghan 618307, China

*Correspondence to*: Huizhi Liu (huizhil@mail.iap.ac.cn)

**Abstract.** Based on eddy covariance (EC) measurements from 2016 to 2020, the impact of sky conditions on net ecosystem productivity (NEP) over Beihai wetland was examined. Sky conditions were classified into sunny, cloudy and overcast skies. On half-hourly timescale, the daytime NEP responds to the changing total photosynthetically active radiation ($PAR_t$) more efficiently under cloudy and overcast conditions than sunny conditions across seasons. Compared with sunny conditions, the apparent quantum yield (α) under overcast (cloudy) conditions increased 342.9% (271.4%) in spring, 17.6% (20.6%) in
summer, 280.0% (230.0%) in autumn and 125.0% (25.0%) in winter, respectively. Unlike the patterns of half-hourly NEP, the daily NEP was significantly lower under overcast conditions than that under cloudy and sunny conditions. And the daily NEP peaked under cloudy skies when the clearness index (CI) fluctuated around 0.3-0.6. Additionally, the ecosystem light use efficiency (LUE) and water use efficiency (WUE) also changed with the variations in sky conditions. The daily LUE and WUE reached their maximum values under overcast (CI: 0-0.2) and cloudy conditions (CI: 0.2-0.4), respectively. NEP was mainly
controlled by the diffuse photosynthetically active radiation ($PAR_d$) and air temperature (Ta), and the direct photosynthetically active radiation ($PAR_b$) had a secondary effect on NEP from half-hourly to monthly timescales. Path analysis revealed that $PAR_b$ and Ta were the main controls affecting NEP under sunny conditions. While under cloudy and overcast conditions, $PAR_d$ was the most responsible for the variations of NEP.

## 1 Introduction

The solar radiation, particularly photosynthetically active radiation (PAR: 400-700nm), provides energy for plant photosynthesis (Park et al., 2018). The amount of global radiation incident at the ground surface varied dramatically, it decreased after 1950s and increased after 1990s, which consequently affecting the eco-physiological processes (Wild, 2009). The total solar radiation can be divided into the diffuse radiation and the direct radiation (Ren et al., 2013). Affected by changes in cloudiness, the total solar radiation and the fraction of diffuse radiation varied significantly across different sky conditions
(Oliphant et al., 2011). This may influence some canopy gas exchange processes, such as net ecosystem exchange (NEE), light





use efficiency (LUE) and water use efficiency (WUE) (Han et al., 2019; Liu et al., 2022; Zhang et al., 2011b), and the terrestrial carbon cycle (Bai et al., 2012; Cheng et al., 2015).

The effects of sky conditions on the canopy photosynthetic characteristics and productivity have been conducted in different ecosystems, including forests (Dengel and Grace, 2010; Gu et al., 1999; Xu et al., 2017), grasslands (Bai et al., 2012; Li et al.,
2020; Wang et al., 2016b), crops (Pearman and Garratt, 2022; Wang et al., 2008), and peatlands (Goodrich et al., 2015; Letts and Lafleur, 2005). Although many studies have been conducted to investigate the canopy photosynthetic characteristics under different sky conditions, there is still no consensus regarding the effects of sky conditions on ecosystem productivity and carbon sequestration capacity. Some research has shown that the cloudy conditions with large fraction of diffuse radiation can enhance the LUE drastically (Bai et al., 2012; Cheng et al., 2015; Kanniah et al., 2012). It also has been reported that the
cloudy conditions can enhance carbon dioxide ($CO_2$) uptake in some forest sites (Cheng et al., 2015; Gu et al., 1999; Xu et al., 2017). However, Alton et al., (2008) found the net primary production was generally reduced when the global radiation declined dramatically under gloomier skies across 38 sites from different ecosystems in FLUXNET. A study conducted over a peatland in Ontario, Canada showed that NEE did not differ between cloudy and clear days (Letts and Lafleur, 2005).

The underlying mechanism for the influence of sky conditions on the canopy photosynthesis and productivity is related to the
diffuse radiation effects. Compared with the direct light, the diffuse light is distributed over more leaves, which can largely reduce photo-saturation and photo-inhibition of the upper canopy (Gu et al., 2002; Knohl and Baldocchi, 2008). In the meanwhile, the diffuse radiation is conducive to reducing water and heat stress of upper canopy, and create favourable conditions for photosynthesis (Cheng et al., 2015; Zhou et al., 2020). Additionally, diffuse radiation can penetrate deeper into the canopy than direct radiation, and illuminate the shaded leaves of lower canopy that may be light-limited under sunny
conditions (Hollinger et al., 1994; Oliphant et al., 2011). There is also a hypothesis that diffuse radiation has higher ratio of blue light. This may stimulate stomatal opening and photochemical reactions, thereby increasing canopy LUE (Urban et al., 2012).

Other environmental variables that vary with sky conditions can also influence plants photosynthesis and the impact of diffuse radiation on ecosystem productivity (Krakauer and Randerson, 2003; Moazenzadeh et al., 2018). It has been reported that the
atmospheric vapor pressure deficit (VPD) and air temperature (Ta) play an important role in the impact of diffuse radiation on the ecosystem productivity (Zhang et al., 2011a), although some research reported that these factors had no significant influence (Jing et al., 2010; Kanniah et al., 2011). The net effect of VPD and Ta associated with sky conditions contributed to photosynthesis enhancement in a mixed deciduous forest (Oliphant et al., 2011). While in a temperate mountain peatland, elevated VPD occurring in clear skies was recognized as a constraint to $CO_2$ uptake (Otieno et al., 2012). Considering the
complex and interactions between multiple environmental factors and diffuse radiation, there are uncertainties in determining the regulations of environmental variables to canopy productivity. Thus, more investigations in different sites are required (Gui et al., 2021; Han et al., 2019; Knohl and Baldocchi, 2008).

The impact of sky conditions on ecosystem productivity has primarily been conducted in forests, while the wetlands have received little attention. As a component of terrestrial ecosystems, wetlands play an important role in the global terrestrial carbon cycle by storing large amounts of carbon, although they only cover 5-8% of the earth's surface (Keddy, 2010). Beihai wetland is an alpine marsh in southwest China, located in the southeast of the Qinghai-Tibet Plateau (QTP). This wetland is permanently inundated, and plants float on the water surface all year round. The height of the plants can up to 2 m. This site situates within the subtropical monsoon climate zone, with large precipitation and high temperature occurring in the wet season (May to October). It has been reported that this wetland has a large $CO_2$ sequestration capacity, with annual mean NEP of 233.8 g C $m^{-2}$ $yr^{-1}$ (Du et al., 2021).

Based on 5-years eddy covariance (EC) datasets of $CO_2$ flux, we explored the effect of sky conditions on NEP over Beihai wetland. The objectives of this paper are to: (1) analyze the diurnal and seasonal variation of NEP under different sky conditions, (2) explore the role of sky conditions in modifying LUE and WUE, (3) investigate how do PAR and environmental variables jointly influence NEP under different sky conditions. Such investigations can not only improve our understanding of the underlying mechanism of ecosystem carbon sequestration in this area, but they can also provide information for improving ecological models.

## 2 Methods

### 2.1 Site description

Field observations were performed at Beihai wetland, Tengchong, Yunnan province of southwest China. The eddy tower changed its location in March 2017, from near the shore (25 ʾ7′ N, 98 ʾ33′ E) to the middle of the wetland (25 ʾ07′N, 98 ʾ33′E, 1728 m a.s.l.). Beihai wetland is located in the Hengduan Mountains, southeast of the Tibetan Plateau. This area is strongly affected by East Asia and South Asia monsoon climate, with distinct wet (May to October) and dry (November to April) seasons. The prevailing wind direction is southwest and northeast. According to the long-term meteorological records (1981-2010) of Tengchong County weather observation station (16.5 km far from the wetland), the annual mean precipitation sum and annual mean Ta are 1532.4 mm and 15.4 ℃, respectively.

Beihai wetland is an alpine marsh wetland, with plants floating over the water surface permanently. The area of Beihai wetland is around 0.46 $km^2$, of which vegetation covers about 0.32 $km^2$ (Zhao and Du, 2014). Around 60% of the wetland surface was covered by vegetation. The dominant plants over this wetland surface were *Cyperus duclouxii E.-G. Camus* and *Oberonia iridifolia Roxb. ex Lindl*, with the maximum height exceeding 2 m. More details are shown in Du et al., (2021).





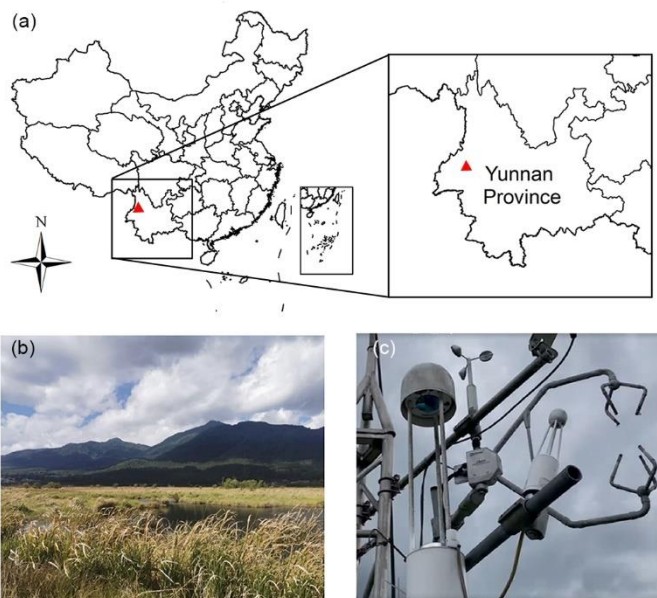


**Figure 1. (a) The location of the experiment site (red triangle) on the map of China. The pictures of (b) field site and (c) EC system.**

## 2.2 Instrumentation

EC system was used to estimate turbulent $CO_2$ flux. The system was consisted of a three-dimensional sonic anemometer (CSAT3A, Campbell, USA), an open-path $CH_4$ analyzer (LI-7700, LI-COR Inc., USA) and an open-path $CO_2/H_2O$ gas

analyzer (LI-7500, LI-COR Inc., USA). The central points of gas analyzers were on the same level as the sonic anemometer. Datalogger (Model CR 3000, Campbell Scientific Inc., USA) sampled the data at 10Hz. At the same time, the radiation components were measured with four radiometers (CNR4; Kipp and Zonen, Delft, the Netherlands). The measurement height of radiation was 1.5 m above both the grass and water surfaces. In the meanwhile, the PAR was monitored at a height of 1.5 m over the grass surface (LI-190SB; LI-COR Inc.). The Ta, relative humidity (HMP45C; Campbell Scientific) and wind speed

(U) (010C; Campbell Scientific) were all measured at three levels (0.65, 1.45, and 3 m above the platform). The wind direction was observed at a height of 3 m above the platform (020C; Campbell Scientific), and the platform was 1.5 m above the wetland surface. The water temperature was observed at 5, 10, 20, 40, and 60 cm below the water surface (109L; Campbell Scientific), while the water surface temperature was calculated by longwave radiation. A tipping bucket rain gauge (52202; Young, USA) was used to measure the precipitation. All meteorological measurements were recorded at half-hourly intervals by dataloggers

(Model CR1000, Campbell Scientific Inc., USA).

## 2. 3 Turbulent flux calculation

Half-hourly turbulent $CO_2$ flux was computed using Eddypro 7.0.4 (LIOR, USA). The post-processing steps included spike filtering (Vickers and Mahrt, 1997), coordinate rotation using double rotation method (Kaimal and Finnigan, 1995), spectral



losses correction (Moncrieff et al., 1997; Moncrieff et al., 2004) and WPL corrections (Webb et al., 1980). The output of
EddyPro included the widely used data flagging system developed by Foken et al. (2005), with data flagged on a 0-1-2 scale.
Data of poor quality with a flag value of 2 and those obtained during rainy days were discarded from the following analysis.
With the quality controls and other system malfunctions (i.e., power failure, instrument malfunctioning, and heavy rain
conditions), 62% of $CO_2$ data were finally reserved.

Missing data were gap-filled using different methods. Linear interpolation was used to fill gaps that were less than 2 h by
calculating an average of the values before and after the data gap. While the data gaps longer than 2 h were filled with the
marginal distribution sample (MDS) based on the replacement of missing values using a time window of several adjacent days
(Falge et al., 2001).

After the gap-filling procedure, half-hourly $CO_2$ flux was partitioned into gross primary production (GPP) and ecosystem
respiration ($R_{eco}$). The value of 20 W m$^{-2}$ of solar radiation ($R_s$) was used as the limit for day and night, and nighttime in this
study referred to $R_s$ less than 20 W m$^{-2}$. The temperature response equation was used to fill the long gaps in nighttime NEE
(Lloyd and Taylor, 1994):

$$NEE_{nighttime} = a \exp (bTs) \tag{1}$$

where $NEE_{nighttime}$ was the NEE in nighttime (μmol m$^{-2}$ s$^{-1}$); Ts was the water temperature at the 5 cm depth (∘C); a and b are
regression parameters. The daytime ecosystem respiration can be calculated from Ts according to Eq. (1), and the daytime
GPP was calculated as following:

$$GPP = R_{eco} - NEE \tag{2}$$

Based on EC measurements, we estimated WUE and LUE. LUE (μmol $CO_2$ μmol PAR$^{-1}$) was calculated as the ratio of GPP
to the total photosynthetically active radiation ($PAR_t$), and WUE (g C mm$^{-1}$ H$_2$O) was calculated as the ratio of GPP to
evapotranspiration (ET):

$$LUE = GPP / PAR_t \tag{3}$$

$$WUE = GPP / ET \tag{4}$$

**2.4 Cloudiness and diffuse radiation**

Due to the lack of direct observations of the cloudiness, the clearness index (CI) was used as a proxy for sky conditions (Gu
et al., 1999). It was defined as the ratio of $R_s$ received above the canopy to the extraterrestrial solar radiation at a hypothetical
horizontal surface ($R_0$). CI was calculated as following:

$$CI = R_s / R_0 \tag{5}$$



$$R_0 = R_{sc} [1+0.033\cos(360d/365)] \sin\beta \tag{6}$$

$$\sin\beta = \sin\varphi\sin\delta + \cos\varphi\cos\delta\cos\omega \tag{7}$$

where $R_{sc}$ is the solar constant (1367 W m$^{-2}$), d is the day of the year, β is the solar elevation angle.

The diffuse and direct photosynthetically active radiation were calculated by $PAR_t$ and the fraction of diffuse photosynthetically active radiation ($k_d$) according to Reindl et al., (1990) as following:

$$k_d = 1.02 - 0.254CI + 0.0123\sin\beta \ (0 \leq CI \leq 0.3) \tag{8}$$

$$k_d = 1.4 - 1.749CI + 0.177\sin\beta \ (0.3 < CI < 0.78) \tag{9}$$

$$k_d = 0.486CI - 0.182\sin\beta \ (CI \geq 0.78) \tag{10}$$

$$PAR_d = k_d \times PAR_t \tag{11}$$

$$PAR_b = PAR_t - PAR_d \tag{12}$$

where $PAR_d$ and $PAR_b$ was the diffuse and the direct photosynthetically active radiation (μmol m$^{-2}$ s$^{-1}$), respectively.

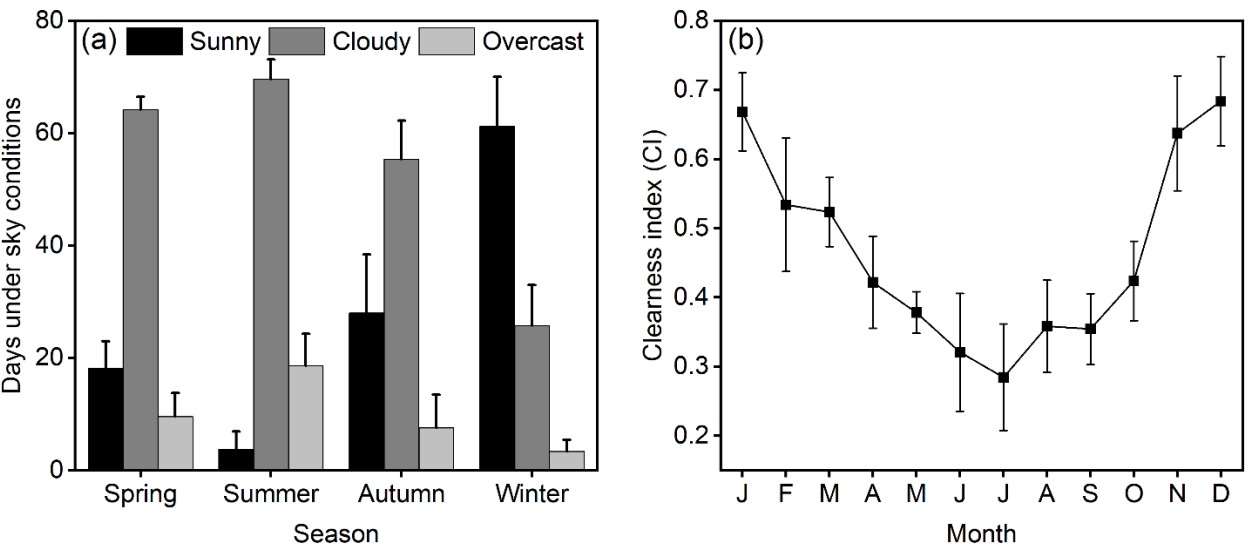

**Figure 2. (a) Days under sunny, cloudy and overcast conditions across seasons, and (b) monthly average clearness index (CI) values**
**(data are shown as monthly average ± standard error) during 2016-2020.**

In most previous studies, sky conditions were only classified as clear or cloudy conditions (Alton, 2008; Li et al., 2020; Xu et al., 2017), and the difference in NEP between cloudy and overcast conditions had not been determined. In this study, we divided sky conditions using daily average CI into sunny ($0.6 \leq CI < 1$), cloudy ($0.2 \leq CI < 0.6$) and overcast ($0 < CI < 0.2$) conditions according to Reindl et al., (1990). Figure 2a showed the days under different sky conditions across seasons during





2016-2020. The weather in spring, summer and autumn were mainly cloudy skies, while the weather in winter was mainly sunny skies. Months during the wet season had the lowest CI values of between 0.2-0.5 (Fig. 2b), indicating frequent cloud cover during this time of the year.

## 2.5 Light response model

In order to investigate how $CO_2$ flux (i.e., NEP = -NEE) response to $PAR_t$ under different sky conditions, an exponential function was chosen to fit the response of NEP to $PAR_t$ (Bassman and Zwier, 1991):

$$\text{NEP} = P_{max} \left[ 1 - \exp\left(-\alpha \times \frac{PAR_t}{P_{max}}\right) \right] - R_d \tag{13}$$

where $\alpha$ is the apparent quantum yield (µmol µmol$^{-1}$), $P_{max}$ is the maximum light saturated NEP (µmol m$^{-2}$ s$^{-1}$), and $R_d$ is the dark respiration (µmol m$^{-2}$ s$^{-1}$).

## 2.6 Statistics analysis

The relationships between NEP and environmental variables, including $PAR_d$, $PAR_b$, Ta, VPD and U, were investigated by partial correlation analysis and stepwise multiple regression analysis on half-hourly, daily and monthly timescales in SPSS (Version 26.0, SPSS Inc., IL, USA).

Path analysis is a mathematical analysis method, which is similar to multiple regression. This research used SPSS AMOS (version 24.0, IBM Inc., USA) to analyze the direct and indirect effects of environmental variables on NEP under different sky conditions. According to the existing knowledge, the structural relationships among the selected variables were established. We used the maximum likelihood estimation method to compute data. In addition to all significant paths ($p < 0.05$), the software could output standardized direct effects (SDE), indirect effects (SIE) and total effects (STE) of each variable. STE is the sum of SDE and SIE. Positive and negative values indicated promotion and inhibition, respectively. And the absolute coefficients values could be used to compare the relative impacts of variables on NEP (Han et al., 2019; Han et al., 2020).

## 3. Results

### 3.1 Environmental conditions and CI

$PAR_t$ exhibited obvious seasonal variation, with the largest values of 828.9 mol mol$^{-1}$ occurring in summer (Fig. 3). $PAR_t$ during the wet season varied within a wide range due to the rainy weather conditions. $PAR_b$ showed similar seasonal variation to $PAR_t$, with a largest value of 524.3 mol mol$^{-1}$. $PAR_d$ exhibited a unimodal variation trend in each year, which reached the maximum of 587.2 mol mol$^{-1}$ in summer. During the wet season, $PAR_t$ was dominated by $PAR_d$. The annual $k_d$ (i.e., the ratio of $PAR_d$ to $PAR_t$) was 0.59 in 2016, 0.57 in 2017, 0.61 in 2018, 0.62 in 2019, 0.63 in 2020, respectively. The maximum Ta occurred in August, with the monthly mean value of about 21.1 ℃. The annual mean Ta during 2016-2020 was 15.5, 15.1,

14.8, 15.2 and 15.3 ℃, respectively, which was below or close to the long-term climate average (15.4 ℃). The water surface

temperature (Tw) followed Ta, and it was larger than Ta. The meteorological variables did not change much from year-to-year

185 except for precipitation. The annual accumulative precipitation from 2016 to 2020 was 1780.2, 1490.3, 1494.4, 1210.4 and

1497.3 mm, respectively. More than 85% of the mean annual precipitation occurred during the wet season. The VPD followed

the annual precipitation distribution, and it generally peaked during spring.

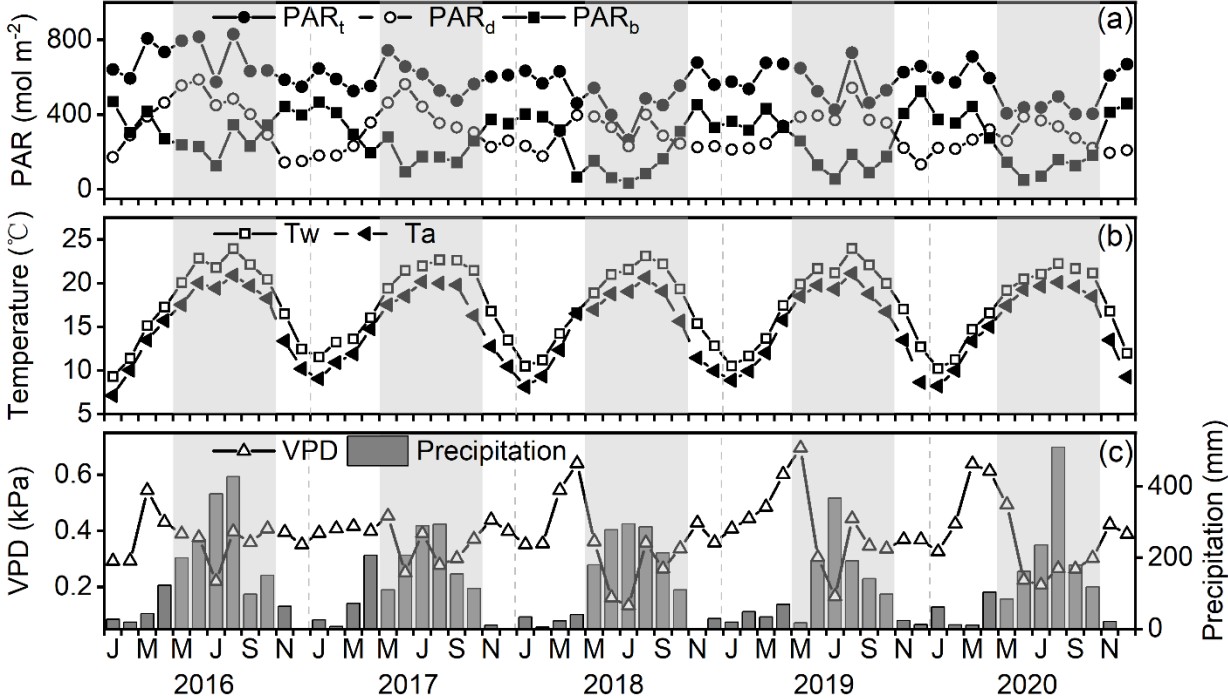

**Figure 3. Seasonal variations of environmental variables. (a) Monthly integrated radiation (PAR$_t$, PAR$_d$ and PAR$_b$ were total, diffuse**

190 **and direct photosynthetically active radiation, respectively), (b) monthly average air temperature (Ta) and water surface**

**temperature (Tw), (c) monthly average atmospheric vapor pressure deficit (VPD) and monthly integrated precipitation during 2016-**

**2020. The shaded area represented wet season (from May to October).**

The variation of cloudiness (indicated by CI) could affect radiation (including PAR$_t$, PAR$_b$ and PAR$_d$), Ta and VPD. PAR$_t$

increased linearly with the increase of CI (Fig. 4a). When CI was low (0-0.3), there was little PAR$_b$ been received above the

195 canopy. When CI exceeded 0.3, the increase in PAR$_b$ was exponentially with respect to CI. The relationship between PAR$_d$

and CI was complex. When the sky was covered by thick clouds (CI: 0-0.2), PAR$_d$ was low owing to high reflection of radiation

by the clouds. As CI increased, PAR$_d$ increased and peaked under thin cloud conditions (CI: 0.4-0.6). When the sky became

clearer (CI > 0.8), PAR$_d$ increased as a consequence of high PAR$_t$. While for Ta and VPD, they both linearly increased with

the increase in CI due to increased incident radiation (Fig. 4b and 4c).





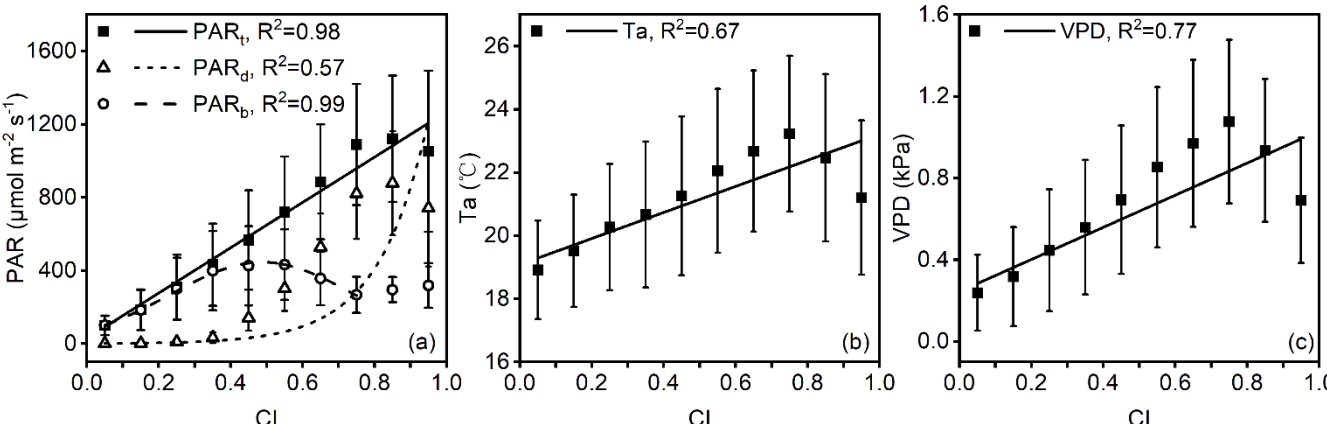

**Figure 4. The relationship between the clearness index (CI) and (a) radiation (PAR$_t$: total photosynthetically active radiation; PAR$_d$: diffuse photosynthetically active radiation; PAR$_b$: direct photosynthetically active radiation), (b) air temperature (Ta), (c) atmospheric vapor pressure deficit (VPD). Data used are half-hourly values during daytime in the wet season from 2016 to 2020. Half-hourly data was bin-averaged by CI increment of 0.1. Bars indicate standard errors.**

## 3.2 Ecosystem photosynthesis under different sky conditions

Light response curves under different sky conditions in four seasons were shown in Fig. 5, and the light response parameters differed under different sky conditions. Under cloudy and sunny conditions, the photo-inhibition phenomena were observed. When PAR$_t$ was greater than 1500 µmol m$^{-2}$ s$^{-1}$, NEP was suppressed primarily because the vegetation absorbed light quantum for photosynthesis was saturated (Fig. 6). Generally, PAR demand over this site was lower than that over an alpine meadow site, where the light saturated when PAR larger than 1800 µmol m$^{-2}$ s$^{-1}$ (Gu et al., 2003).

The light response curve parameters were listed in Table 1 and Table 2. Seasonally, the apparent quantum yield (α) was higher under overcast and cloudy skies than that under sunny skies in all seasons. The increase of α under cloudy conditions was 271.4% in spring, 20.6% in summer, 230.0% in autumn and 25.0% in winter, respectively; and the increase of α under overcast conditions attained 342.9% in spring, 17.6% in summer, 280.0% in autumn and 125.0% in winter, respectively (Table 1). Although α was the greatest in summer, higher increase of α occurred in spring and lower increase occurring in summer. As the sky became cloudy, the P$_{max}$ also became larger, except in summer and winter. Compared with sunny conditions, P$_{max}$ under cloudy conditions increased by 2.2% in spring, -2.5% in summer, 7.3% in autumn and -36.6% in winter, respectively; and P$_{max}$ under overcast conditions increased by 39.8% in spring, -8.5% in summer, 24.4% in autumn, and -72.0% in winter, respectively.

Annually, α and P$_{max}$ were both larger under cloudy skies than that under sunny skies (Table 2). Compared with sunny conditions, α under cloudy conditions increased by 150.0% in 2016, 118.8% in 2017, 146.2% in 2018, 92.9% in 2019 and 175.0% in 2020, respectively; and P$_{max}$ under cloudy conditions increased 21.9% in 2016, 31.4% in 2017, 26.1% in 2018, 85.9% in 2019 and 74.6% in 2020, respectively. Overall, compared with sunny conditions, α and P$_{max}$ under overcast conditions





increased by 200.0% and 73.9% on average; α and P$_{max}$ under cloudy conditions increased by 120.0% and 60.1% on average,
respectively.



**Figure 5. Light response curves of net ecosystem productivity (NEP) to the total photosynthetically active radiation (PAR$_t$) on the daytime half-hourly basis across seasons during 2016-2020.**





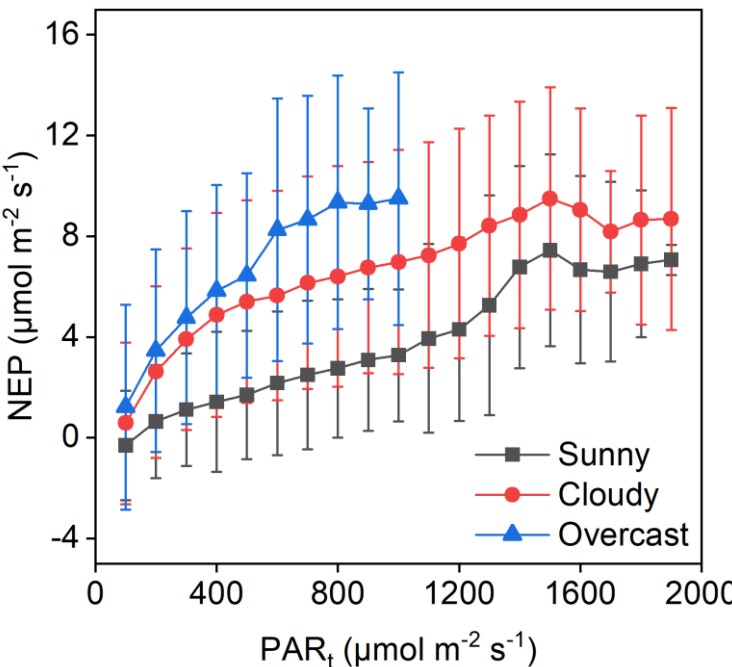

**Figure 6. Relationship between half-hourly net ecosystem productivity (NEP) and the total photosynthetically active radiation (PAR$_t$) during daytime under sunny, cloudy and overcast conditions. NEP was bin-averaged by PAR$_t$ increment of 100 µmol m$^{-2}$ s$^{-1}$, and bars indicate standard errors.**

**Table 1. The light-response curve parameters under different sky conditions in four seasons from 2016 to 2020.**

| Season | Weather | α | P$_{max}$ | R$_d$ |
|--------|---------|---|-----------|-------|
|  |  | (µmol µmol$^{-1}$) | (µmol m$^{-2}$ s$^{-1}$) | (µmol m$^{-2}$ s$^{-1}$) |
| Spring | Sunny | 0.007 | 6.86 | 1.21 |
|  | Cloudy | 0.026 | 7.01 | 1.13 |
|  | Overcast | 0.031 | 9.59 | 2.31 |
| Summer | Sunny | 0.034 | 13.10 | 2.74 |
|  | Cloudy | 0.041 | 12.77 | 1.76 |
|  | Overcast | 0.040 | 11.98 | 0.88 |
| Autumn | Sunny | 0.010 | 8.92 | 0.43 |
|  | Cloudy | 0.033 | 9.57 | 1.69 |
|  | Overcast | 0.038 | 11.1 | 2.33 |
| Winter | Sunny | 0.004 | 7.82 | 0.92 |
|  | Cloudy | 0.005 | 4.96 | 0.78 |
|  | Overcast | 0.009 | 2.19 | 0.69 |





**Table 2. The light-response curve parameters under different sky conditions in five years during 2016 to 2020.**

| Year | Weather | $\alpha$ | $P_{max}$ | $R_d$ |
|------|---------|----------|-----------|-------|
| | | ($\mu mol\ \mu mol^{-1}$) | ($\mu mol\ m^{-2}\ s^{-1}$) | ($\mu mol\ m^{-2}\ s^{-1}$) |
| 2016 | Sunny | 0.012 | 8.93 | 0.32 |
| | Cloudy | 0.030 | 10.89 | 0.18 |
| | Overcast | 0.026 | 6.20 | 0.58 |
| 2017 | Sunny | 0.016 | 10.98 | 0.78 |
| | Cloudy | 0.035 | 14.43 | 1.13 |
| | Overcast | 0.032 | 7.51 | 1.83 |
| 2018 | Sunny | 0.013 | 6.51 | 1.22 |
| | Cloudy | 0.032 | 8.21 | 1.55 |
| | Overcast | 0.044 | 6.20 | 0.82 |
| 2019 | Sunny | 0.014 | 5.25 | 0.95 |
| | Cloudy | 0.027 | 9.76 | 1.60 |
| | Overcast | 0.047 | 12.55 | 1.68 |
| 2020 | Sunny | 0.012 | 4.52 | 1.18 |
| | Cloudy | 0.033 | 7.89 | 1.72 |
| | Overcast | 0.027 | 8.36 | 0.95 |
| All year | Sunny | 0.010 | 5.87 | 0.87 |
| | Cloudy | 0.022 | 9.40 | 0.93 |
| | Overcast | 0.030 | 10.21 | 0.90 |

### 3.3 Diurnal and daily patterns of NEP under different sky conditions

NEP showed a distinct diurnal variation under different sky conditions, and $CO_2$ uptake generally reached its maximum value during the midday (Fig. 7a-d). The maximum value was 11.6 $\mu mol\ m^{-2}\ s^{-1}$, occurring in summer under sunny conditions, which was similar to the results in other wetlands (Anderson et al., 2016; Fortuniak et al., 2017; Yu et al., 2020). The difference in diurnal patterns of NEP under cloudy and sunny conditions was not marked, while NEP under sunny and cloudy conditions was evidently greater than that under overcast conditions. The depression of $CO_2$ uptake at noon, which was found in some grasslands (Wang et al., 2016a; Wang et al., 2017), was not significantly for the diurnal patterns of NEP over study site. There were some reasons, on the one hand, the weather in spring, summer and autumn was mainly cloudy days (Fig. 2) and the radiation on cloudy days was relatively low (Fig. 4). Although the weather in winter was dominated by sunny skies, the total radiation was comparatively low during this period. On the other hand, the depression of NEP in some grasslands was also caused by the soil water stress under high-level radiation (Fu et al., 2006; Li et al., 2005), which did not occur in this wetland.





In the meanwhile, NEP under different skies on daily timescale across seasons was also compared (Fig. 7e-dh). The daytime NEP was positive in all seasons. The strongest daytime $CO_2$ uptake occurred in summer, while the weakest daytime $CO_2$ uptake occurred in winter. And the daytime NEP was even close to 0 under overcast conditions in winter. There was larger daytime NEP under sunny or cloudy conditions than that under overcast conditions. Seasonally, compared with sunny days,

the daytime NEP in overcast days decreased by 25.0% in spring, 54.5% in summer, 28.6% in autumn and 97.4% in winter, respectively; and the daytime NEP in cloudy days increased by 27.3%, -19.0%, 34.5% and -28.3%, respectively, in four seasons. After considering the contribution of nighttime ecosystem respiration, the daily NEP was relatively smaller than the daytime NEP in all seasons across sky conditions, and it might even become a carbon source. Daily NEP varied with the variation of CI, and peaked when CI was 0.3-0.6 (Fig. 8a).

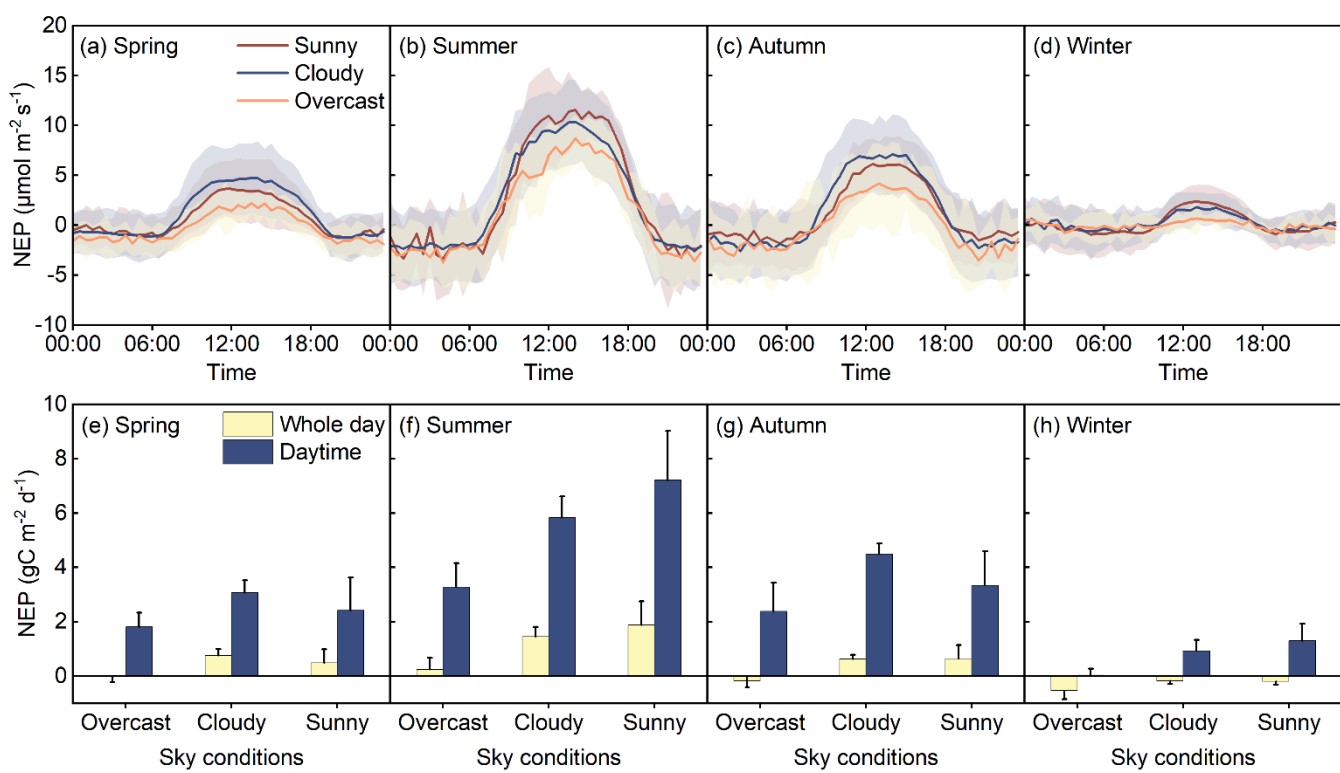

**Figure. 7. Diurnal variations of net ecosystem productivity (NEP) under different sky conditions across seasons from 2016 to 2020 (a-d), all data were averaged by the half-hourly data under the same sky condition in the same season. Daily integrated net ecosystem productivity (NEP) and daytime NEP ($R_s > 20$ W m$^{-2}$) under different sky conditions across seasons during 2016-2020 (e-h).**

### 3.4 The relationships among LUE, WUE and CI

Cloudiness could affect ecosystem LUE and WUE through $k_d$ (Aires et al., 2008; Min, 2005; Oliphant et al., 2011). As CI increased, LUE decreased as a result of the increase in $k_d$ (Fig. 8). LUE reached its maximum under overcast conditions (CI:





0-0.2). LUE increased by 298.7% when sky condition changed from sunny to overcast, and by 158.6% when sky conditions changed from cloudy to overcast. WUE was critical for quantifying the relationship between water consumption and ecosystem photosynthetic production. The variation of $PAR_d$ under different sky conditions might alter the balance between transpiration

and photosynthesis, thereby altering WUE (Liu et al., 2022). WUE varied with CI, and peaked under cloudy conditions when CI was 0.2-0.4. The average WUE during 2016-2020 was $1.34 \pm 0.21$ g C $mm^{-1}$ $H_2O$, which was within the range obtained over other sites (0.65-5.4 g C $kg^{-1}$ $H_2O$) (Brümmer et al., 2012; Yu et al., 2008). The difference in WUE among sites was mainly related to the different geographical, climate and environmental conditions (Liu et al., 2022).

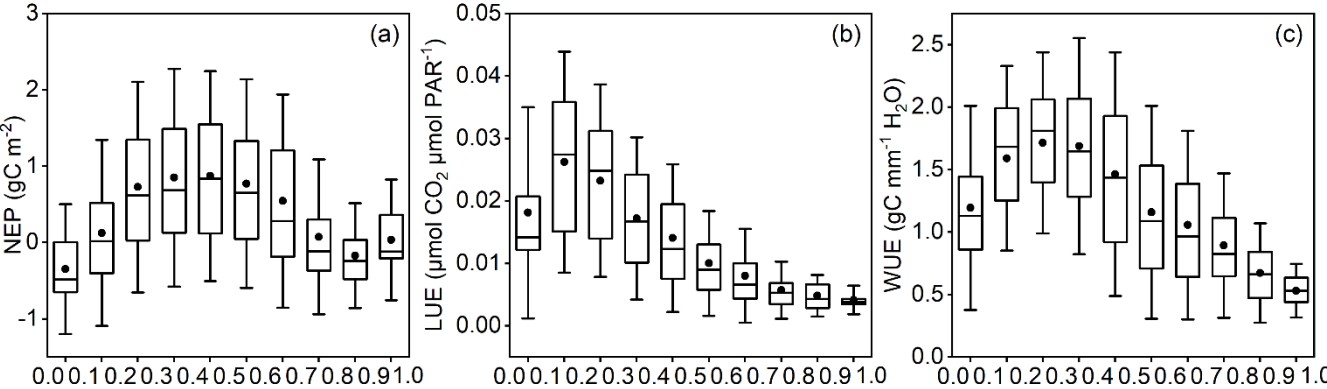

**Figure 8. Changes in daily (a) net ecosystem productivity (NEP), (b) light use efficiency (LUE), and (c) water use efficiency (WUE) during daytime with the clearness index (CI) during 2016-2020.**

### 3.5 Influence of environmental variables on NEP

Correlations between environmental variables and NEP were analyzed by the partial correlation analysis and stepwise multiple regression analysis on half-hourly, daily and monthly timescales. The partial correlation coefficients between NEP and some

variables (i.e., $PAR_d$, $PAR_b$, and Ta) were positive, while the coefficients between NEP and VPD were negative across different timescale (Fig. 9a). NEP was mainly controlled by $PAR_d$ and Ta, and $PAR_b$ had a secondary effect on NEP from half-hourly to monthly scales. On half-hourly timescale, $PAR_d$ played the dominant role in regulating the variations in NEP, and it could explain 29.4% of NEP variation (Fig. 9b). As the timescales was extended to daily and monthly timescales, the relative contribution of $PAR_d$ to NEP decreased (22.6% on daily timescale and 31.2% on monthly timescale). $PAR_b$ determined 15.0%,

10.4%, and 12.1% variations of NEP on half-hourly, daily and monthly timescale, respectively. In the meanwhile, the relative contribution of Ta to NEP varied considerably across different timescales, ranging from 18.4% (half-hourly timescales) to 50.3% (monthly timescales). The increase in VPD inhibited NEP, and 1.9%-16.8% of the variation of NEP was explicated by VPD from half-hourly to monthly timescales.





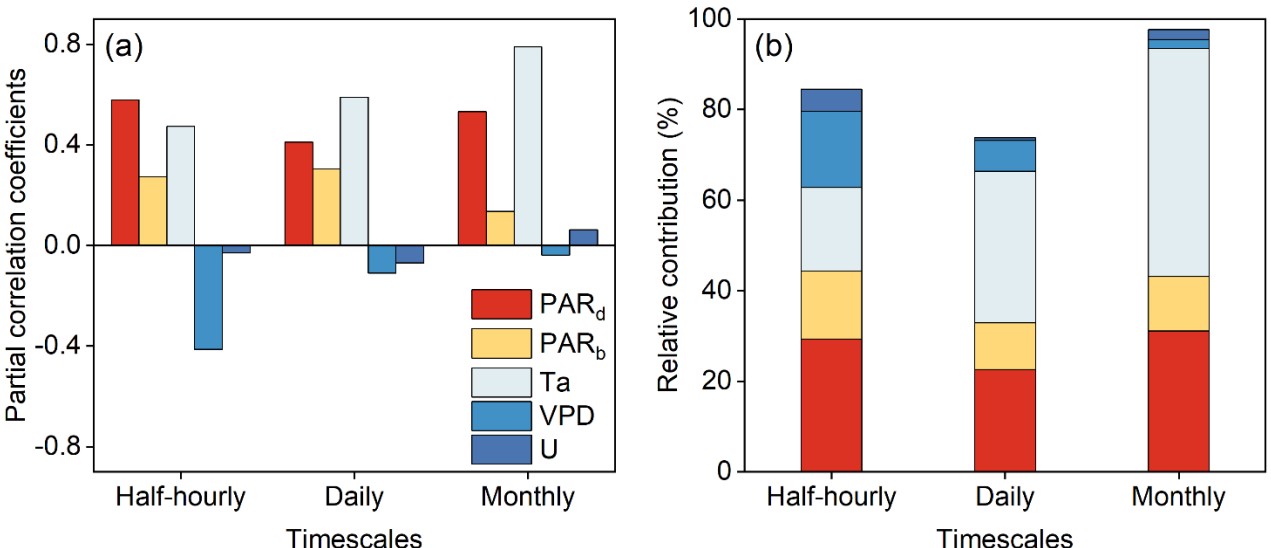

Figure 9. (a) Responses of the integrated net ecosystem productivity (NEP) to environmental factors from half-hourly to monthly timescales. (b) The relative contribution of environmental factors to NEP on different timescales.

In order to compare the relative contribution of environmental variables to NEP under different sky conditions, path analysis was performed in four seasons (Fig. 10). Our result showed that solar radiation affected NEP mainly through SDE, while the effect through SIE was relatively negligible (Table 3). In spring, STE of $PAR_b$ on NEP was significantly strong under different sky conditions (Fig. 11). In summer, STE of $PAR_d$ was the strongest under overcast and cloudy conditions, while STE of Ta and $PAR_b$ was stronger than other variables under sunny conditions. In autumn, STE of environmental variables on NEP was similar to that in summer. In winter, STE of $PAR_b$ on NEP was the main promotion under sunny and cloudy conditions, while VPD was the main controlling factors of NEP under overcast conditions. Overall, STE of environmental variables on NEP under sunny conditions was ordered as follows: $PAR_b$ > Ta > $PAR_d$ > VPD. The order under cloudy conditions was sorted by: $PAR_d$ > Ta > $PAR_b$ > VPD. Under overcast conditions, the order of variables was: $PAR_d$ > Ta > VPD > $PAR_b$.





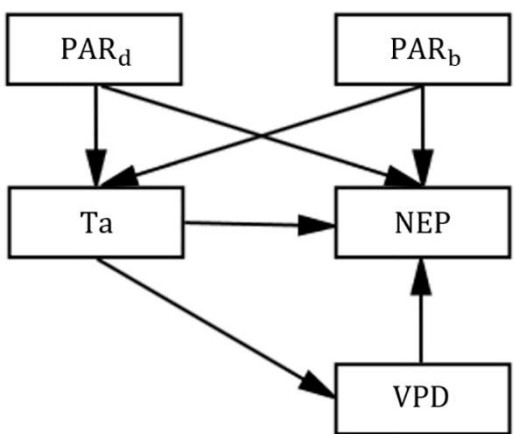

**Figure 10. Conceptual path analysis diagram of effects of environmental variables on net ecosystem productivity (NEP). PAR$_d$, PAR$_b$, Ta and VPD indicated the direct photosynthetically active radiation, diffuse photosynthetically active radiation, air temperature, and atmospheric vapor pressure deficit, respectively.**

**Table 3. Standardized total effects (STE), standardized direct effects (SDE), and standardized indirect effects (SIE) of diffuse photosynthetically active radiation (PAR$_d$) and direct photosynthetically active radiation (PAR$_b$) on daily net ecosystem productivity (NEP) under different sky conditions in four seasons during 2016-2020.**

| Season | Weather | PAR$_d$ | | | PAR$_b$ | | |
|---|---|---|---|---|---|---|---|
| | | STE | SDE | SIE | STE | SDE | SIE |
| Spring | Sunny | 0.423 | 0.260 | 0.164 | 0.533 | 0.450 | 0.083 |
| | Cloudy | 0.269 | 0.281 | -0.012 | 0.375 | 0.291 | 0.084 |
| | Overcast | -0.111 | -0.312 | 0.202 | 0323 | 0.447 | -0.124 |
| Summer | Sunny | 0.081 | 0.077 | 0.004 | 0.286 | 0.442 | -0.156 |
| | Cloudy | 0.320 | 0.327 | -0.007 | 0.188 | 0.220 | -0.032 |
| | Overcast | 0.609 | 0.629 | -0.020 | 0.042 | 0.039 | 0.003 |
| Autumn | Sunny | 0.011 | -0.092 | 0.102 | 0.349 | 0.152 | 0.197 |
| | Cloudy | 0.279 | 0.176 | 0.103 | 0.249 | 0.261 | -0.012 |
| | Overcast | 0.641 | 0.560 | 0.082 | 0.044 | 0.133 | -0.089 |
| Winter | Sunny | 0.310 | 0.323 | -0.013 | 0.352 | 0.366 | -0.013 |
| | Cloudy | 0.169 | 0.170 | -0.001 | 0.329 | 0.330 | -0.001 |
| | Overcast | 0.108 | 0.149 | -0.041 | 0.216 | 0.150 | 0.066 |




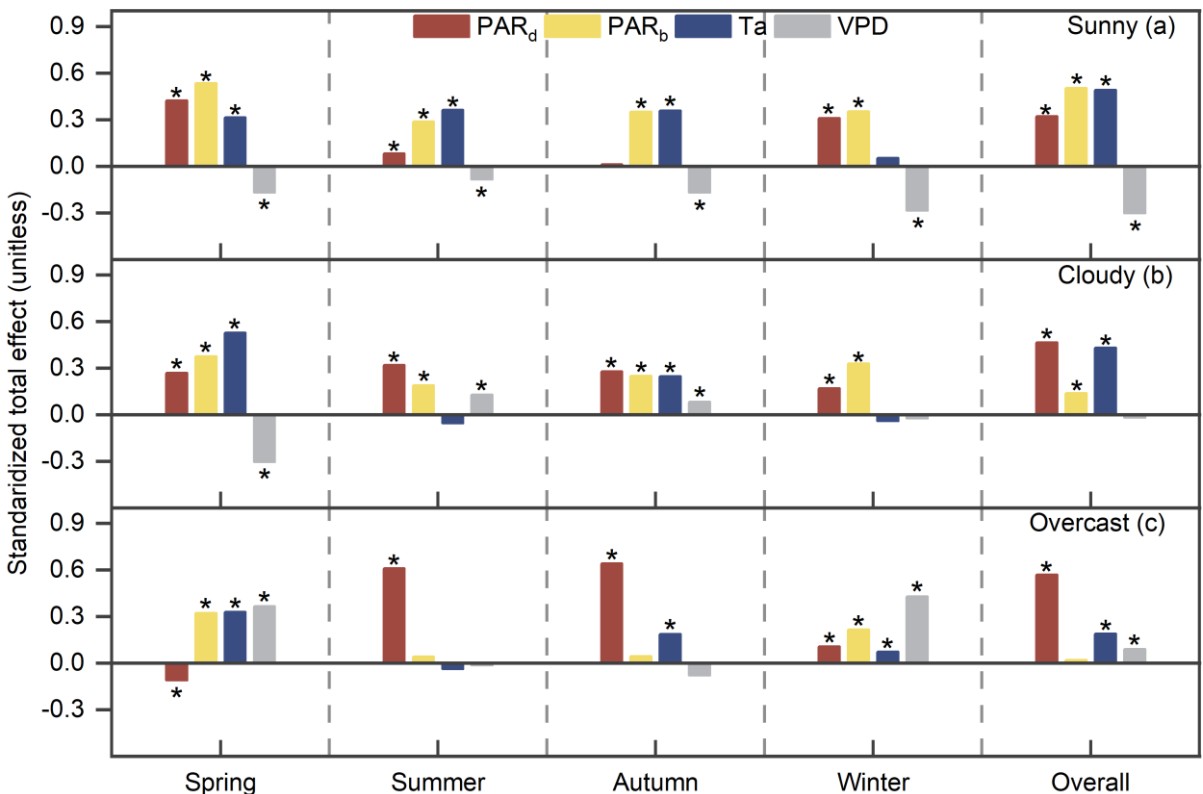

**Figure 11. Standardized total effects (STE) of diffuse photosynthetically active radiation (PARd), direct photosynthetically active radiation (PARb), air temperature (Ta), and atmospheric vapor pressure deficit (VPD) on daytime net ecosystem productivity at half-hourly scale under (a) sunny, (b) cloudy, and (c) overcast sky conditions across seasons. STE in overall was obtained by all data selected under given sky conditions. Asterisks (∗) represent the significance was at the level of p ≤ 0.05.**

## 4. Discussion

### 4.1 The effects of diffuse radiation on ecosystem photosynthetic characteristics

Sky conditions had a significant effect on ecosystem photosynthesis (Fig. 5). The light response curves under different sky conditions showed similar trends, and the curves under overcast conditions increased more steeply with $PAR_t$. This result was consistent with previous research (Li et al., 2020; Oliphant et al., 2011; Williams et al., 2014). The response of photosynthesis to sky conditions varied across seasons (Table 1). We concluded that the photosynthetic parameters (i.e., $\alpha$ and $P_{max}$) were greater under overcast and cloudy conditions than that under sunny conditions, and the increase of $\alpha$ under cloudy and overcast conditions was greater in spring while smaller in summer. The difference in the increase of $\alpha$ might be related to the variation of $k_d$. Although $k_d$ in summer was the larger than other seasons, the increase of $k_d$ under cloudy and overcast conditions was greater in spring and smaller in summer. In spring, compared with sunny conditions, $k_d$ under cloudy and overcast conditions





increased by 188.5% and 310.4%, respectively. While in summer, $k_d$ only increased by 26.3% and 53.2% under cloudy and
overcast conditions, respectively.

The increase of α and $P_{max}$ under cloudy and overcast exhibited obvious year-to-year variation. While there was no clear
relationship between the annual variation of photosynthetic parameters and the annual variation of $k_d$. This was probably
because these parameters were influenced by a mixture of other environmental variables, and the effect of $k_d$ might be
overlapped by other factors (Stoy et al., 2005; Wu et al., 2017). During the observation period, there was distinct year-to-year
variation in annual total precipitation, while the year-to-year variations of other environmental variables was not significant.
The total precipitation was the lowest in 2019 (1210.4 mm), which was 21.0% less than the long-term climate average (1532.4
mm). The total precipitation was the highest in 2016 (1780.2 mm), which was 16.2% more than the long-term climate average.
The larger difference of annual precipitation and the uneven distribution of precipitation may be responsible for the year-to-
year variation of ecosystem photosynthetic capacity (Han et al., 2019; Rocha et al., 2004).

Our result showed that LUE was also affected by diffuse radiation, and it differed under different skies. Compared with sunny
conditions, higher $k_d$ under cloudy and overcast conditions improved LUE, and LUE peaked under overcast skies (CI: 0-0.2),
which was similar to previous research (Alton et al., 2007; Kanniah et al., 2013). It has been found that under strong UV
radiation, plant would regulate pigmentation and enzyme mechanisms, and reduce photosynthesis to protect themselves from
high levels of radiation  (Correia et al., 2005; Ekelund, 2000; Li et al., 2010). Therefore, the canopy LUE under sunny
conditions was low. While under cloudy and overcast conditions, increased $k_d$ could promote the canopy photosynthesis by
reducing the stress of high radiation of upper canopy and increasing the light of lower shaded layers (Dengel and Grace, 2010;
Kanniah et al., 2013; Urban et al., 2007). Additionally, changes in environmental variables were also responsible for the
enhancement in canopy LUE under cloudy skies. Kannian et al. 2013 found that canopy LUE was not sensitive to sky
conditions when VPD and Ta were relatively high, while LUE could be elevated under favourable environmental condition
(i.e., low VPD and Ta).

### 4.2 Effect of diffuse radiation on daily NEP

To investigate the effect of $PAR_d$ on ecosystem productivity, daily NEP under different sky conditions in different seasons
was compared. Unlike the half-hourly NEP, the majority of daytime and all-day NEP on daily timescale was significantly
larger under cloudy and sunny conditions than that under overcast conditions. Compared with sunny conditions, the daytime
NEP on daily scale decreased by 55.0% under overcast conditions, and increased by 88.2% under cloudy conditions. The
difference in the patterns of NEP on half-hourly and daily timescales might be related to the discrepancy of timescales (Han
et al., 2019; Stoy et al., 2005; Wu et al., 2017). As the timescale was extended from half-hourly to daily timescale, the impact
of Ta on NEP increased (Fig. 9). Diurnal variation of Ta and other environmental factors could affect the integrated NEP
throughout a whole day. Daily NEP varied with CI and peaked when CI fluctuated around 0.3-0.6, which lagged slightly when




PAR$_d$ peaked. This CI rang was comparable to that in other sites (CI: 0.4-0.7) (Bai et al., 2012; Gu et al., 1999; Jing et al., 2010).

Our result indicated that increased PAR$_d$ could enhance NEP through its diffuse fertilizer effect, which was similar to other studies (Alton et al., 2007; Knohl and Baldocchi, 2008; Mercado et al., 2009). While in some research, higher production rate was observed under clear sky conditions (Alton, 2008; Han et al., 2019; Kanniah et al., 2013). There were some reasons. First, the quality of solar radiation might be more critical than the quality of radiation for NEP in some ecosystems. As the cloudiness increased, changes in photosynthesis depended on the balance between the reduction of PAR$_t$ and the enhancement of PAR$_d$ (Mercado et al., 2009). The canopy photosynthesis began to decrease when PAR$_t$ on the sunlit leaves fell below the light saturation point, and the reduction in photosynthesis could not be compensated by the enhanced photosynthesis of shaded leaves, which benefited from increased PAR$_d$ (Misson et al., 2005). Kanniah et al., (2013) reported a reduction of 26% in GPP when PAR decreased by 63% under thick clouds. Alton (2008) also reported a general decrease of 60-80% of the net primary production when the global radiation declined dramatically across 38 sites from different ecosystems in FLUXNET. Second, the canopy structures, such as leaf area index and the height of the canopy, were also critical factors in determining the role of PAR$_d$ on ecosystem productivity (Greenwald et al., 2006; Kanniah et al., 2012). A study conducted over a peatland in Canada showed there was little difference in mean NEE across all ranges of CI above 0.3, owing to the low leaf area index (LAI) and small stature of the canopy (Letts and Lafleur, 2005). The small LAI and stature of the canopy tended to relatively low light extinction and small limited shading of leaves in the lower canopy (Frolking et al., 2002). While those canopies with large LAI generally contributed to promote NEE under cloudy conditions (Bai et al., 2012; Park et al., 2018). Other canopy characteristics, such as foliar N concentration and leaf mass per area could affect photosynthesis by altering the light uniformity over the canopy (Heskel et al., 2012; Wright et al., 2004). In addition, the leaf inclination angle and leaf optical properties (transmittance and reflectance) also change the effect of PAR$_d$ on NEP, and thereby influence ecosystem productivity (Alton et al., 2007; Williams et al., 2014). Finally, the light saturation points of plants could influence the impact of PAR$_d$ on NEP. It had been found that mosses had very low PAR saturation levels, ranging from 50 μmol m$^{-2}$ s$^{-1}$ to 300 μmol m$^{-2}$ s$^{-1}$ (Goulden and Crill, 1997), which was much lower than the PAR saturation levels of grasslands and the observed wetland (~1500 μmol m$^{-2}$ s$^{-1}$) (Gu et al., 2003; Wang et al., 2016b). Therefore, the ground cover of moss was almost always light saturated under most sky conditions, and enhanced PAR$_d$ under cloudy skies was unlikely to increase NEE in any part of the canopy compared to sunny conditions.

Compared with sunny conditions, WUE was found to be higher under cloudy conditions. As observed in this study, increased WUE under cloudy conditions might be responsible for greater $CO_2$ uptake from the atmosphere for a given amount of water loss. ET could continue to increase linearly with the available energy, while the photosynthesis might saturate under sunny conditions. Under cloudy conditions, the increase in cloudiness reduced the ET by reducing the amount of total solar radiation, while the increased PAR$_d$ was conducive to enhancing photosynthesis and GPP (Freedman et al., 2001). This result was similar





to the result that was found in other sites (Rocha et al., 2004; Min, 2005; Kanniah et al., 2013). Liu et al., (2022) investigated WUE of six plantations under different sky conditions, and found that the low VPD and high canopy conductance increased carbon assimilation but reduced water loss under cloudy conditions, which was conducive to WUE enhancement.

385 **4.3 Environment regulations on NEP under different sky conditions**

Different sky conditions induced variations of radiation composition and other environmental factors, leading to a confounding effect on NEP. The radiation played the dominant role in regulating the variations of NEP under different skies, which was also found in other sites (Goodrich et al., 2015; Han et al., 2020; Letts and Lafleur, 2005). Under sunny conditions, $PAR_b$ and Ta had significant promotion effects on NEP in four seasons (Fig. 11). STE of $PAR_d$ followed by $PAR_b$ and Ta, representing 390 a little promotion on NEP. While under cloudy and overcast conditions, $PAR_d$ was the main controlling factor of NEP, and Ta had a secondary effect on NEP. Our path analysis results showed that the regulation of radiation on NEP is mainly through direct effect, and the indirect effect was less pronounced.

Ta was treated as an important factor affecting NEP in our site, and it played a general positive role in promoting NEP across different sky conditions. The increase in Ta could improve the enzyme activity, electron transfer efficiency for photosynthesis 395 and leaf carboxylation rates, which the plants could use to promote transpiration rates and photosynthetic rates (Huang et al., 2019; Oliphant et al., 2011; Son et al., 2014). The promotion effect of Ta on canopy productivity was also reported in other studies (Han et al., 2020; Zhang et al., 2006). However, extremely high Ta might limit photosynthesis. Elevated Ta could not only accelerate leaf ageing and increase leaf thickness, but also lead to an increase in VPD and the closure of plant stomata (Huang et al., 2019; Williams et al., 2012). In addition to being a constraining factor for NEP, Ta also played a mediating role 400 in the influence of $PAR_d$ on NEP. Some research had reported that there was an optimum temperature range for the strongest effect of $PAR_d$ on NEP (Zhang et al., 2020). Obviously, the influence of each environmental factor on NEP needed to be understood in the context of other factors.

VPD was another critical environmental variable for NEP, and the increase in VPD generally led to a decrease in photosynthesis (Gui et al., 2021). On the one hand, leaf cells were strongly influenced by the balance of intra- and extra-leaf 405 water vapor pressure, and the leaf stomata tended to close when VPD increased (Goodrich et al., 2015; Zhou et al., 2013). This might reduce the intercellular and chloroplasts $CO_2$ concentration, which reduced the photosynthetic capacity (Urban et al., 2007). On the other hand, high VPD might limit the enzymatic processes and constrain biochemical capacity, as well as increase the mesophyll resistance (Flexas et al., 2012; Niinemets et al., 2006; Sage, 2002), which could restrict NEP. Our study also found VPD had an inhibition effect on NEP under sunny conditions. While under cloudy and overcast skies, the negative 410 effect of VPD on NEP decreased, and VPD even promoted NEP under overcast conditions. This kind of change in the effect of VPD on NEP under different sky conditions was also found in the study by Han et al., (2020). It's mainly because VPD was low under overcast skies (mean value of 0.12 ±0.06 kPa) (Fig. 4), the ecosystem was limited by light conditions rather than



water conditions. Therefore, the limitation of NEP by VPD could be reduced. A study conducted in a tropical rainforest showed that when VPD increased, ecosystem photosynthesis decreased more significantly in dry years than that in wet years (Zhang

et al., 2011b).

## 5 Conclusions

Effects of sky conditions on NEP over an alpine marsh wetland in southwest China were evaluated by using EC data during 2016-2020. We found the response of daytime NEP to the changing $PAR_t$ was suppressed when $PAR_t$ was greater than 1500 $\mu$mol m$^{-2}$ s$^{-1}$. The daytime NEP on half-hourly scale was generally more positive under overcast and cloudy skies than under

sunny skies, and the light response parameters (i.e., $\alpha$ and $P_{max}$) were also greater under cloudy and overcast conditions in different seasons. While the daily NEP in cloudy and sunny days was significantly larger than that in overcast days. NEP peaked under cloudy conditions with a CI of 0.3-0.6. The difference in the patterns of NEP on half-hourly and daily timescales might be related to the discrepancy of timescales. In the meanwhile, LUE and WUE also changed with the variation of sky conditions. LUE and WUE reached their maximum value under overcast (CI: 0-0.2) and cloudy conditions (CI: 0.2-0.4),

respectively. It meant that cloudy sky conditions could enhance LUE and WUE over Beihai wetland.

The partial correlation analysis and stepwise multiple regression analysis were used to investigate the relative importance of environmental variables on NEP across different timescales. $PAR_d$ and Ta were the main controls of NEP from half-hourly to monthly timescales, and $PAR_b$ had a secondary impact on NEP. On half-hourly scale, $PAR_d$, $PAR_b$ and Ta could explain 29.4%, 15.0% and 18.4% of NEP variation, respectively. As the timescales was extended to daily and monthly timescales, the relative

contribution of $PAR_d$ and $PAR_b$ to NEP decreased and the relative contribution of Ta increased. The increase in VPD inhibited NEP, and 1.9%-16.8% of the variation of NEP was explicated by VPD from half-hourly to monthly timescales. Overall, under sunny conditions, NEP was mainly controlled by $PAR_b$ and Ta. While under cloudy and overcast conditions, $PAR_d$ contributed most to the variation of NEP.

## Code availability

The codes used in the preparation of this paper are available upon request from the authors.

## Data availability

Please contact the corresponding author to access data.



## Author contributions

Yamei Shao and Yang Liu performed the measurements; Jihua Sun contributed data; Yamei Shao analyzed the data and wrote
the manuscript draft; Huizhi Liu, Qun Du and Yaohui Li reviewed and edited the manuscript.

## Competing interests

The authors declare that they have no conflict of interest.

## Acknowledgements

We acknowledge the support of National Natural Science Foundation of China (Grant No.91937301), the Second Tibetan
Plateau Scientific Expedition and Research (STEP) program (Grant No. 2019QZKK0105), and National Natural Science
Foundation of China (Grant No. 41975017, 41905010). We were also very grateful to the Senior engineer (Liu Dingting) of
Tengchong County Meteorological Bureau for his help in the maintenance of the measurement instruments.

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
