# Peer review of "Impact of sky conditions on net ecosystem productivity of a "floating blanket" wetland in southwest China"

_Biogeosciences, 2022_

## Author Comment (AC1)

**Response to RC1:**

The manuscript presents an empirical analysis of the relationship between cloudiness and NEP. From the upstart I see a serious problem in the approach. Specifically, with regards to the variables they choose to analyze.

**Response:** We would like to thank anonymous Referee #1 for his valuable comments and questions. It is very helpful to improve this paper. Responses to all the points raised by the referee are in the following:

First, why look at effects on NEP? Radiation (par) has direct effects on photosynthesis and thus, GPP, and is not expected to affect Respiration. Any effect on respiration is indirect, either through increased sugar transport to the roots, or through increase temperature. Furthermore, in the analysis of WUE, why use ET and not transpiration? especially in a flooded wetland, the direct evaporation from the water surface and saturate soil related to radiation in a very different way that transpiration and has nothing to do with carbon uptake so it should not be included in WUE.

**Response:** Radiation can influence photosynthesis in both direct and indirect ways, in the meanwhile, it can also affect respiration through temperature. This implies radiation plays an important role in regulating NEP. In addition, previous researches have investigated the effects of sky conditions or diffuse radiation on GPP, while their effects on NEP have less been studied, and there is still great uncertainty about the impact on NEP over different ecosystems. Therefore, we would like to investigate effects on NEP.

The vegetation and water cover distributed over the Beihai wetland heterogeneously, and the fraction of vegetation to water surface changes with season, which poses a great challenge for the partitioning of ET. We have not partitioned ET, which will be tried in the future.

Second, the key driver of this analysis, cloudiness, was not observed, instead it is derived indirectly. More serious is that direct and diffuse PAR fractions where not observed (a sensor for direct/diffuse shortwave radiation exist and is not very expensive). This study modelled direct/diffuse PAR from the calculated cloudiness index and observed total PAR using the empirical equations (eq 8-12) by Reindl et al 1990. These equations where parameterized in coastal northern (US and Europe) locations, which is a very different than the climate type, latitude and elevation of the current study site (this site is at least 1500 m above the highest site of Reindl). I therefore question the accuracy of the Reindl equations to this site.

**Response:** Both direct measurement and model estimation are main methods for calculating cloudiness and direct/diffuse PAR. We don't have the direct measurements of cloudiness and direct/diffuse PAR over the study site currently and will try supplement them in the future. There are several empirical equations for calculating the direct/diffuse PAR, and the method of Reindl et al., (1990) is widely used domestically and internationally over different sites due to its simple form. These studied sites range in elevation from a few tense of meters to more than 1000 meters. Moreover, the empirical method by Reindl et al., (1990) shows good performance in simulating diffuse PAR over site at a latitude and climate zone similar to ours. Therefore, we

believe the empirical equation developed by Reindl et al., (1990) can be used in this study.

Finally, all environmental variables covary, with strong diurnal and seasonal cycles. For example, if you repeat the analysis from table 1 but based on time of day (e.g., compare 7-9 am to 12-2 pm) in the summer you will find very different alpha, $P_{max}$ and Rd. The point here is that the affects you attribute to more diffuse radiation, could be actually the effects of lower temperature or higher humidity. Your analysis approach does very little to disentangle the covarying drivers of transpiration and photosynthesis. Your path analysis confirms it (without actually solving the problem).

Especially for a single-site study, which is not generalizable from the start, getting more depth in the analysis of the hypothetical effects and linking the observations better to current models for the effects of direct/diffuse radiation is critical, and missing from this study.

**Response:** We use the variance inflation factor (VIF) and path analysis method to determine the multi-collinearity issue. Serious multi-collinearity issue among variables is considered to exist when VIF is greater than 10, while VIFs under different sky conditions are all less than 10 in this study. Consequently, we can conclude that no serious multi-collinearity relation was found among these meteorological variables. In addition, path analysis is effective for evaluating data in which independence among variables is not certain, because it can exclude potential relationships between independent variables. It is suitable for quantifying the direct and indirect effects of environmental variables on NEP. The values of alpha, $P_{max}$ and $R_d$ may be different if we use data from different time periods, however, this may not change the result that diffuse radiation exerts greater impact on NEP. We have considered the effect of temperature and VPD on NEP, while path analysis shows the impact of radiation is stronger. More discussions about analysis of the hypothetical effects and the linkage between the observations and current models will be added in the revised manuscript.

Other, easier to address comments:
How are you calculating Reco (eq. 2)?

**Response:** NEE at night is nighttime Reco, because there is no photosynthesis during nighttime. Then we calculate the regression coefficients between nighttime NEE and Tw according to Eq. (1). We assume the daytime temperature response of Re is the same as the nighttime one, and then we can calculate daytime Reco based on the regression equation of nighttime NEE and nighttime Tw.

Table 1 and table 2 do not indicate any form of uncertainty (goodness of fit? RMSE?)

**Response:** Sorry we missed the model uncertainty, and we will add the uncertainty in table 1 and table 2 in the revised manuscript.

"Please contact the corresponding author to access data" is not a valid data availability statement. Please publish the half-hourly dataset of meteorological and flux observations. Preferably through ChinaFlux.

**Response:** We are glad to share our data to other researchers who might be interested, and we are working on an application to join ChinaFlux.

---

## Author Comment (AC2)

**Response to RC2:**

Based on the EC observation data from 2016 to 2020, this paper analyzes the impact of weather conditions on the net ecosystem productivity (NEP) of Beihai Wetland. The purpose is to study the influence of weather conditions on NEP, LUE and WUE, as well as the control of scattered radiation and other environmental factors on NEP under different weather conditions.

It is found that the influence of weather conditions on NEP is different on different time scales. On the half-hour scale, the daytime response of NEP to PAR is stronger under cloudy conditions than under sunny conditions. In addition, results show that the daily LUE and WUE change with the cloud, and both LUE and WUE reach the maximum under cloudy conditions.

Using EC observations, this paper analyzes the impact of weather conditions on the net ecosystem productivity of the North Sea wetlands. The authors ultimately want to study the effect of weather conditions on NEP, LUE, and WUE, as well as the control of scattered radiation and other environmental factors on NEP under different weather conditions. Overall, the author can control the full paper, with clever ideas, clear, smooth writing, and attractive titles. It is a rare observational research paper. I suggest publishing after minor revisions.

**Response:** We would like to thank anonymous Referee #2 for his valuable comments on this manuscript. It is very helpful to improve this paper. Responses to all the points raised by the referee are in the following:

There is a small suggestion, you can seriously think about it. In Section 4.2, the impact of scattered radiation on NEP will also be controlled by the vegetation characteristics of the region, are not introduced and analyzed in detail in this paper. I hope you can add some more in detailed expression.

**Response:** We agree that the vegetation characteristics could be one of the factors influencing the impact of diffuse radiation on NEP. However, we lack the vegetation amount measurement, while the vegetation indices (LAI, etc) are greatly affected in this area due to the long rainy season, so the effects of these vegetation features are not analyzed in detail. We will try supplement vegetation amount measurement in the future, and will add more discussions about the impact of some vegetation characteristics in the revised manuscript.